# Derivation and Validation of a New Visceral Adiposity Index for Predicting Short-Term Mortality of Patients with Acute Ischemic Stroke in a Chinese Population

**DOI:** 10.3390/brainsci13020297

**Published:** 2023-02-10

**Authors:** Yuhong Chang, Lulu Zhang, Yidan Li, Dapeng Wang, Qi Fang, Xiang Tang

**Affiliations:** 1Department of Neurology, The First Affiliated Hospital of Soochow University, Suzhou 215006, China; 2Department of Neurology, Dushu Lake Hospital Affiliated to Soochow University, Suzhou 215000, China

**Keywords:** visceral adiposity index, acute ischemic stroke, short-term mortality, predictive model, metabolic disorders

## Abstract

The visceral adiposity index (VAI) is related to the occurrence of various cardiometabolic diseases, atherosclerosis, and stroke. However, few studies have analyzed the impact on the short-term prognosis of stroke. We assessed the effect of VAI on short-term prognoses in patients with acute ischemic stroke through a retrospective cohort study of 225 patients with acute stroke who were admitted to the neurological intensive care unit of our hospital. We collected metabolic indicators (blood pressure, fasting glucose, lipids), National Institutes of Health Stroke Scale (NIHSS) scores, symptomatic intracranial hemorrhage, and other disease evaluation indicators on 197 patients who were screened for inclusion. VAI was calculated by using baseline data (sex, height, weight, waist circumference (WC)). We assessed functional recovery according to modified Rankin scale scores after 90 days. The receiver operating characteristic (ROC) curve was used to calculate the VAI cutoff value that affects short-term outcomes. A nomogram that can predict the risk of short-term mortality in patients with acute ischemic stroke was drawn. In total, 28 patients died within 90 days. Those patients had higher VAI (*p =* 0.000), higher triglyceride (TG) (*p =* 0.020) and NIHSS scores (*p =* 0.000), and lower high-density lipoprotein cholesterol (HDL-C) (*p =* 0.000) than patients who survived. VAI had higher predictive value of short-term mortality than did body mass index (BMI), body fat mass index (BFMI), and WC. VAI and NIHSS scores were independent risk factors for the short-term mortality of patients with stroke. Patients with a VAI > 2.355 had a higher risk of short-term mortality. VAI has a predictive value higher than that of traditional metabolic indicators such as BMI, BFMI, and WC. The nomogram, composed of NIHSS, VAI, HDL-C, and TG, may predict the short-term mortality of cerebral infarction patients.

## 1. Introduction

Stroke is one of the leading causes of chronic disability and death worldwide, and its incidence gradually increases with age. A screening study showed that the stroke mortality rate among Chinese residents was between 3% and 25% [1].

The incidence of metabolic disorders is also high in patients with stroke, with one study showing that approximately 62% of patients with ischemic stroke have metabolic syndrome [2]. Obesity is an important causal factor of metabolic disorders. In recent years, global obesity rates have increased [3]. It is also increasingly recognized that fat distribution plays a more important role in obesity-induced complications related to metabolic disorders than total fat content [4,5,6,7]. Visceral adipose tissue (VAT) has been shown to be associated with insulin resistance and the risk of cardiovascular and cerebrovascular disease [5,7,8,9,10], whereas subcutaneous adipose tissue (SAT) appears to be more protective. Therefore, indicators that reflect the differences in fat distribution across individuals have been developed and used gradually. The visceral adiposity index (VAI) calculation model, established by Amato et al. in 2010, is a sex-specific indicator based on waist circumference (WC), body mass index (BMI), triglycerides, and high-density lipoprotein cholesterol [8], which is used for predicting VAT mass and function. VAI is verified as an independent risk factor for cardiovascular events. In follow-up studies, it has also been proven that VAI is closely related to the occurrence of metabolic diseases, cardiovascular diseases, atherosclerosis, and asymptomatic cerebral infarction [11,12,13,14,15,16,17]. A positive correlation between VAI and the incidence of stroke has also been found in some studies conducted in the Chinese population [18,19]. A study on the distribution of visceral fat and the prognosis of acute ischemic stroke found that low visceral abdominal fat proportion is associated with a favorable and excellent outcome in patients with acute ischemic stroke who were treated with intravenous thrombolysis. Moreover, better clinical outcomes in obese patients were also associated with a lower proportion of VAT [20]. In some domestic studies, a VAI calculation model more suitable for Chinese people (CVAI) has also been established and verified as better at predicting the occurrence of metabolic syndrome, hypertension, and diabetes in the Chinese population [21]. Although many studies on the relationship between VAI and stroke incidence have been published, there is little research on the correlation between VAI and the prognosis of patients with cerebral infarction.

At present, it is well known that obesity is an important predictor of cardiovascular and cerebrovascular diseases, but more and more studies have found that obesity is negatively correlated with a poor prognosis of stroke, which is called the “obesity paradox” [22,23]. Subsequent more-in-depth studies have found that obesity is not associated with a poor early prognosis of stroke, but it has a protective effect on a long-term prognosis [24,25]. This paradox remains controversial because of the absence of randomized trials, the retrospective nature of most studies, the assessment of obesity with the body mass index (BMI), and the nonlinear relationship between BMI and outcomes [26]. In further exploration of the obesity paradox, some studies have used VAI to evaluate the obesity status of individuals. An investigation of VAI and the risk of all-cause mortality in elderly people that was based on data provided by NHANES in the United States found that there was a J-shaped relationship between VAI and all-cause mortality [27]. Meanwhile, there is also no consensus on the relationship between metabolic indicators, such as TG, TC, HDL-C, and LDL-C, and the prognosis of patients with acute stroke. One study found that low HDL-C levels were associated with an increased risk of stroke mortality [28], while another showed that low HDL-C levels were only an independent predictor of stroke severity, not stroke prognosis [29]. Two studies found that low TG levels were positively associated with poor stroke outcomes, while one found that TC levels could also predict stroke outcomes [30], and the other contradicted this conclusion [29]. In addition, studies on the correlation between LDL-C and stroke prognosis found that low LDL-C did not show a protective effect on the risk of death after stroke [31] or had no correlation with stroke prognosis [29]. Therefore, the relationship between overweight, obesity, and metabolic disorders on one hand and the short-term prognosis of patients with acute cerebral infarction on the other also remains controversial. Metabolic syndrome, a well-known indicator of obesity and metabolic disorders, has been repeatedly confirmed in previous clinical studies to be associated with the risk of early death in patients with ischemic stroke [32]. Therefore, whether the VAI calculated using WC, TC, HDL-C, and BMI as baseline data is also associated with poor prognosis in patients with ischemic stroke is worth exploring.

The purpose of this study was to analyze the association of obesity and metabolic-related indicators, including VAI, BMI, and body fat mass index (BFMI), with short-term mortality risk in patients with acute ischemic stroke and compare the predictive value of VAI and other metabolic-related indicators on the short-term mortality risk of patients with acute ischemic stroke in our hospital. Finally, we aimed to draw a nomogram to establish a prediction model of metabolic-related indicators for the short-term mortality of patients with acute ischemic stroke.

## 2. Materials and Methods

### 2.1. Participant Enrollment

We enrolled 252 patients with ischemic cerebrovascular disease who were admitted to the intensive care unit of the neurology department of the First Affiliated Hospital of Soochow University from October 2018 to April 2021. Inclusion criteria were as follows: (1) aged older than 18 years; (2) acute stroke occurred within 72 h of onset; and (3) stroke diagnosed by brain computed tomography (CT) or magnetic resonance imaging (MRI). The main exclusion criteria were as follows: (1) a modified Rankin scale (mRs) score greater than two points before the onset of the disease; (2) a combination with severe craniocerebral injury or intracranial tumor; (3) a combination with diseases that may interfere with the evaluation of the results, such as cardiovascular disease, tumor, liver, and kidney failure, etc.; (4) patients who undergo other surgical treatment; and (5) an inability to participate in the 3-month follow-up owing to various reasons. According to the exclusion criteria, we excluded 40 patients, including those with a history of cerebral infarction with mRs ≥ two points (*n =* 5), those who received another surgical treatment for the disease (*n =* 1), those with severe craniocerebral injury or intracranial tumor and diseases that may interfere with the evaluation of the results (*n =* 10), and those who were lost to follow-up and who refused to participate in the 90-day follow-up (*n =* 18). Finally, we admitted 197 patients with acute ischemic stroke who met the inclusion criteria (122 men and 75 women) and analyzed the data collected (Figure 1).

### 2.2. Data Collection

Basic information, such as age, sex, BMI, WC, blood pressure, stroke type (TOAST classification), stroke severity (based on the National Institutes of Health Stroke Scale; NIHSS), past history (high blood pressure, diabetes, coronary heart disease, smoking, etc.), mRs scores before onset, whether thrombolysis or endovascular treatment had been received, and other laboratory data (high-density lipoprotein cholesterol, low-density lipoprotein cholesterol, triglycerides, blood sugar, glycosylated hemoglobin, etc.), was collected upon admission. The VAI was calculated by using the following formula [9]:

Men: VAI *=* [WC/(39.68 + BMI × 1.88)] × (TG/1.03) × (1.31/HDL-C)

Women: VAI *=* [WC/(36.58 + BMI × 1.89)] × (TG/10.81) × (1.52/HDL-C)

The CVAI was calculated by using the following formula [16]:

Men: CVAI *=* −267.93 + 0.68 × age + 0.03 × BMI + 4.00 × WC + 22.00 × Log_10_TG − 16.32 × HDL-C

Women: CVAI *=* −187.32 + 1.71 × age + 4.23 × BMI + 1.12 × WC + 39.76 × Log_10_TG − 11.66 × HDL-C

A bioelectrical impedance analysis was performed by using Bodystat to assess the body fat mass index (BFMI). The presence or absence of symptomatic intracranial hemorrhage during the hospital stay was recorded at discharge. Finally, we conducted the follow-up visit at 3 months after discharge, by telephone. The items included a 90-day mRs score, 90-day recurrence, and 90-day mortality.

### 2.3. Statistical Analysis

The VAI of the enrolled patients was calculated and grouped according to the tertile method, and the NIHSS scores, symptomatic intracranial hemorrhage, and other clinical baseline characteristics of the three groups of patients were statistically analyzed. The VAI, CVAI, BMI, BFMI, WC, and other clinical baseline characteristics of the patients were statistically analyzed on the basis of whether the patients died within 90 days. Measurement data were expressed as mean ± standard deviation (x ± s), using a *t* test. Categorical data were expressed by assignment (%), using the Chi-square test or Mann–Whitney U test. If VAI had a significant effect on 90-day mortality, then the cutoff point of VAI for 90-day mortality was evaluated by using the receiver operating characteristic (ROC) curve. A multiple logistic regression analysis was used to determine whether VAI was independently associated with short-term mortality in patients with acute ischemic stroke admitted to the neurological intensive care unit. Finally, we drew a nomogram of short-term mortality risk factors in patients with acute ischemic stroke admitted to the neurological intensive care unit according to the above analysis results, using fivefold cross-validation to internally validate the results. All analyses were performed using SPSS version 21 (IBM, Armonk, NY, USA). Here, *p* < 0.05 was considered to be statistically significant.

## 3. Results

The baseline characteristics of the enrolled participants are summarized in Table 1. Male patients (*n =* 122) were grouped according to the tertile method after calculating VAI as follows: Q1 (*n =* 41): VAI < 1.12, Q2 (*n =* 41): 1.90 < VAI < 1.12, and Q3 (*n =* 40): VAI > 1.90. Among them, the mean systolic blood pressure of the three groups of patients was Q1: 154.41 ± 18.12, Q2: 144.07 ± 20.53, and Q3: 143.45 ± 20.12, and patients with larger VAI had lower systolic blood pressure (*p =* 0.020). The mean values of high-density lipoprotein cholesterol (HDL-C) among the three groups were Q1: 1.23 ± 0.28, Q2: 0.92 ± 0.21, and Q3: 0.76 ± 0.18, with the patients’ VAI negatively correlated with HDL-C values (*p =* 0.000), which were the same as the systolic blood pressure values. In contrast to low-density lipoprotein cholesterol (LDL-C) and triglycerides (TG), the mean TG values of the three groups of patients were Q1: 0.84 ± 0.20, Q2: 1.13 ± 0.23, and Q3: 1.75 ± 0.51, while the mean LDL-C was Q1: 2.30 ± 0.82, Q2: 3.00 ± 1.34, and Q3: 2.73 ± 1.14; with the increase in the VAI values, the mean values of TG and LDL-C increased (*p =* 0.000, *p =* 0.021). The groupings of female patients (*n =* 75) were: Q1 (*n =* 25): VAI < 1.55, Q2 (*n =* 25): 2.40 < VAI < 1.55, and Q3 (*n =* 25): VAI > 2.40. Among them, the mean BFMI values of the three groups of patients were Q1: 10.22 ± 2.74, Q2: 9.78 ± 2.48, and Q3: 8.06 ± 2.14, while the mean HDL-C values were Q1: 1.30 ± 0.28, Q2: 1.00 ± 1.46, and Q3: 0.90 ± 0.18. It can be seen that the larger the VAI values, the lower the BFMI and HDL-C values of the female patients (*p =* 0.007, *p =* 0.000). Conversely, as VAI increased, LDL-C, TG, and WC increased in female patients (*p =* 0.043, *p =* 0.000, *p =* 0.006). According to the above analysis results, male patients and female patients showed similarities in the relationships between TG, HDL-C, LDL-C, and VAI. We also distinguished the TOAST classification of male and female patients in each VAI interval. Among male patients, the numbers of large-artery atherosclerosis (LAA) for the Q1, Q2, and Q3 groups were 21, 27, and 26, and that of cardioembolism (SE) was 11, 4, and 3. The numbers of small-vessel occlusion lacunar (SAA) for these three groups were 8, 7, and 10, and those of stroke of another undetermined toiology were one, two, and one. In female patients, the numbers of LAA for the Q1, Q2, and Q3 groups were 11, 16, and 15, and those of SE were seven, six, and four. There were six, three, and five cases of SAA and zero, zero, and one cases of stroke of another undetermined toiology. In male and female patients, the difference in VAI was not statistically related to the proportion of different TOAST subtypes. This suggests that higher VAI did not mean more large-artery atherosclerotic strokes or more cardioembolic strokes in our study.

As shown in Table 2, among the 197 patients with acute ischemic stroke, 28 patients died within 90 days. Among them, 19 patients died from neurological causes, and nine patients died from non-neurological causes. On the basis of whether the patients died within 90 days, we analyzed the relationship between various factors and the 90-day death of patients with acute ischemic stroke. The results of the analysis showed that the 28 patients who died had higher NIHSS scores (18.18 ± 9.21, *p =* 0.000), triglycerides (1.43 ± 0.50, *p =* 0.020), and VAI (2.93 ± 1.35, *p =* 0.000) and had more low HDL-C values (0.78 ± 0.18, *p =* 0.000). There was no significant difference in age, BMI, CVAI, WC, BFMI, TOAST classification, or other indicators between patients who died and surviving patients.

We calculated and plotted the ROC curves of VAI, CVAI, WC, BMI, BFMI, and 90-day death in patients with acute stroke. As shown in Figure 2 and Table 3, the area under the curve (95% confidence interval, CI) of VAI was 0.789 (0.695–0.883) among all patients, which was larger than other indicators. The VAI cutoff value was 2.355 (sensitivity, 71%; specificity, 84%). After stratifying by sex, we found that, similar to the calculations for all patients, both male patients and female patients had a larger area under the curve for VAI than for other obesity-related measures, meaning that compared to CVAI, WC, BMI, and BFMI, VAI had a higher predictive value for short-term mortality in patients with cerebral infarction. In male patients, the area under the curve (95% CI) for VAI was 0.762 (0.631–0.893), with a cutoff value of 2.202 (sensitivity, 71%; specificity, 80%). In female patients, the area under the curve of VAI (95% CI) was 0.809 (0.675–0.943), and the cutoff value was 2.765 (sensitivity, 79%; specificity, 85%). After separately drawing the ROC curve of VAI and NIHSS scores, we found that if VAI and NIHSS scores are included at the same time, the area under the curve can reach 0.857, which is higher than the single VAI or NIHSS scores. According to the above data, we found that the predictive value of VAI for women may be higher than that for men.

Finally, we included the indicators that were significantly associated with 90-day death in patients with cerebral infarction in univariate analysis (triglycerides, NIHSS score, and VAI) and included age and sex into a multivariate logistic regression analysis (Table 4). The results showed that VAI (odds ratio *=* 5.944, 95% CI *=* 2.752–12.837, *t =* 1.782, *p =* 0.000) and NIHSS scores (odds ratio *=* 1.136, 95% CI *=* 1.068–1.207, *t =* 0.127, *p =* 0.000) were independent risk factors for 90-day mortality risk in patients with acute ischemic stroke admitted to the neurological intensive care unit. On the basis of these results, we initially established a predictive model for 90-day mortality in patients with acute ischemic stroke, which was expressed in the form of a nomogram (Figure 3). The drawn nomogram was internally validated by fivefold cross-validation, where the areas under the curve of the five validations were 0.848, 0.956, 0.907, 0.819, and 0.765, with an average of 0.856 (95% CI: 0.776–0.935).

## 4. Discussion

In this study, we found that among patients with acute ischemic stroke who were admitted to the neurological intensive care unit, VAI was positively correlated with 90-day death, and this factor could be used as an independent risk factor for predicting death, with a certain predictive value. Moreover, the NIHSS score at admission was also a valuable short-term predictor of mortality, which is consistent with previous studies [33,34]. In our study, we did not find in our hospital that CVAI has a higher predictive value than VAI for short-term mortality in patients with acute ischemic stroke. Correspondingly, traditional obesity assessment indicators, such as BMI, are also less predictive. This may be because BMI evaluates only the overall level of obesity, without distinguishing between fat and muscle mass and without highlighting the distribution of fat; moreover, it is affected by age, race, and sex differences [35]. This also proves that metabolic disorders have a greater impact on the prognosis of patients with cerebrovascular disease than on that of patients with obesity.

The VAI is a relatively novel calculation model, and visceral fat has gradually received attention in recent years. In studies of neurological diseases, in addition to stroke, VAT and VAI were positively correlated with cerebral small-vessel disease and binge eating behavior with caudal gray matter density in the anterior cingulate cortex [36,37]. Visceral fat is also considered a potential link to cognitive decline in healthy older adults and to the development of Alzheimer’s disease [38,39]. In other severe systemic diseases, the VAI has shown predictive value for poor prognosis. In 2020, Engin Celik et al. conducted a study on VAI and the prognoses of patients with endometrial cancer and confirmed that VAI is an important and reliable predictor of endometrial cancer prognosis as well as an independent risk factor affecting the survival rate of patients with endometrial cancer [40]. In other studies, visceral adiposity has been found to be significantly associated with prognosis and survival in hepatocellular carcinoma and pancreatic cancer as well as with the severity of COVID-19 and disease progression [41,42,43]. Although these studies used imaging techniques or body composition analysis to calculate the visceral adiposity content or visceral adiposity area rather than VAI, they further demonstrated the important value of visceral fat in various serious systemic diseases and provided a certain reference for future research on the relationship between VAI and the occurrence, development, recurrence, and prognosis of these diseases.

The novelty of this study is that we for the first time evaluated the association of VAI with the short-term mortality of acute ischemic stroke. In addition to some of the abovementioned studies, past studies have confirmed that VAI can predict the poor prognosis of patients with non-ST-segment elevation acute coronary syndrome and type 2 diabetes treated with percutaneous coronary intervention [44]. Some studies have also found that VAI is closely related to all-cause death in the population and patients with chronic kidney disease [27,45]. In terms of cerebrovascular diseases, many studies have confirmed that VAI is related to the occurrence of atherosclerosis and stroke [16,17,18,19,46], but there is little study on the correlation between VAI and the prognosis of ischemic cerebrovascular disease. VAT was also found to be associated with a poor prognosis of acute ischemic cerebrovascular disease in this research in 2019 [20], which used abdominal computed tomography to measure the thickness of visceral fat, while our study used a formula to calculate the VAI.

Why is the VAI able to predict the short-term mortality risk for patients with acute ischemic stroke who are admitted to the neurological intensive care unit? Although its pathological mechanism is still unclear, several plausible mechanisms can be considered. First, triglycerides and high-density lipoprotein cholesterol included in the calculation formula of VAI have been confirmed to be related to the risk of mortality in patients with stroke, in several previous studies [28,47,48,49]. Higher triglycerides and lower HDL cholesterol were also found in our study to be associated with a higher risk of early death. VAI, as a calculation model for combining these indicators and for the coefficient correction of these indicators, may have a higher correlation than these single biochemical indicators thanks to the synergistic effect of multiple factors [8]. In addition, the expression of T-lymphocyte and macrophage markers was increased in visceral adipose tissue compared with subcutaneous adipose tissue, which was associated with a greater tendency to release more proinflammatory cytokines (e.g., IL-6, IL-8, TNF-α) [50,51]. During the pathological process after the onset of ischemic stroke, the release of these proinflammatory cytokines can aggravate the neuroinflammatory response after stroke [52,53], stimulate the release of neurotoxic mediators, destroy the blood–brain barrier, and induce brain cell edema and endothelial cell apoptosis after stroke, thereby aggravating brain damage in patients with stroke and worsening their prognosis [52,54,55,56,57,58,59]. At the same time, VAT also has a more significant correlation with systemic inflammation-related indicators, such as the neutrophil–lymphocyte ratio and C-reactive protein [60,61]. Meanwhile, one study has shown that persistent systemic inflammation in patients with severe stroke is associated with a poor short-term prognosis and higher 1-month mortality [62]. The above speculation further implies that the VAI has better predictive prospects than traditional obesity indicators.

VAI has some advantages in clinical applications. First, compared with traditional indicators, VAI can better reflect individual adipose distribution and blood lipid status, and its predictive value for metabolic syndrome has also been confirmed several times [63,64,65], which means that the VAI can better predict the individual’s metabolic risk. Second, the traditional measurement of visceral adipose distribution requires methods, such as CT, ultrasound, or bioelectrical impedance, while the calculation of VAI needs height, weight, high-density cholesterol, and triglycerides, which is simpler and easier than traditional measurements and can be widely used and promoted in clinical practice. It is well known that metabolic disorders are significantly related to the onset and prognosis of cardiovascular and cerebrovascular diseases as well as some systemic diseases. The VAI, as a simple index for assessing visceral adipose distribution and predicting metabolic disorders, is worth looking forward to in clinical applications. In more depth, an indispensable future research direction would be to more precisely assess the visceral adiposity index to predict mortality in acute lacunar and nonlacunar ischemic strokes. Compared with macrovascular disease, acute small-vessel ischemic stroke has mildly clinical symptoms and a lower mortality rate, but follow-ups show that it still has a considerable risk of recurrence and early death [66]. The pathophysiological mechanism of acute small-vessel ischemic stroke is also different from other types of cerebral infarction, while the possible pathogenesis of lacunar infarction includes hypertensive arteriosclerosis, atherosclerotic plaque, and an inflammatory response (TNF-, IL-6) [67], which have been shown to be associated with visceral adiposity [16,52,53,68].

In our study, CVAI was not found to be associated with short-term mortality in patients with ischemic stroke in our hospital, and its predictive value was inferior to VAI. However, in some other studies, it was found that CVAI was significantly associated with the incidence of carotid atherosclerosis and coronary heart disease in the Chinese population [68]. In the research that proposed the CVAI computational model, it was also confirmed that this model can better predict the occurrence of hypertension and diabetes [21]. This may be because the original VAI formula can better calculate the visceral fat content of an individual and has higher generality. However, owing to the racial differences in obesity and fat distribution, we hope that the Chinese-specific visceral adiposity content calculation model can be improved in subsequent studies with larger sample sizes.

This study has several limitations. First, because this is a single-center study, the data suffer from some selection bias. Second, the sample size of this study is small, and there is a large sampling error. Third, this study is a retrospective cohort study, and there may have been information bias in the process of data collection. Finally, we did not use more data to justify the findings. Therefore, our research results may not have promotion value. In the follow-up study, we need to collect multicenter data, expand the sample size, and conduct certain prospective studies to confirm our findings.

## 5. Conclusions

VAI is independently associated with the short-term mortality of patients with acute ischemic stroke admitted to the neurological intensive care unit. Higher VAI levels suggest a higher short-term mortality risk. VAI can be used as a predictor of short-term mortality risk in patients with acute ischemic stroke, and its predictive value is significantly higher than that of traditional obesity indicators such as BMI, WC, and BFMI. The nomogram, composed of NIHSS, VAI, HDL-C, and TG, may predict the short-term mortality of cerebral infarction patients. In clinical applications, VAI is an index that can effectively evaluate metabolic conditions and is easy to collect and calculate, which is conducive to clinical promotion.

Moreover, the results of this study imply that body fat distribution and metabolic factors play a greater role in stroke prognosis than obesity alone. Patients with metabolic problems before stroke onset are more likely to suffer poorer outcomes. The higher predictive value of VAI than that of other metabolic indicators also provides a new research direction for improving the poor prognosis of acute ischemic stroke.

Therefore, we suggest evaluating the VAI in newly admitted patients with acute stroke in clinical practice, so as to more effectively analyze the risk of poor prognosis and actively intervene.

## Figures and Tables

**Figure 1 brainsci-13-00297-f001:**
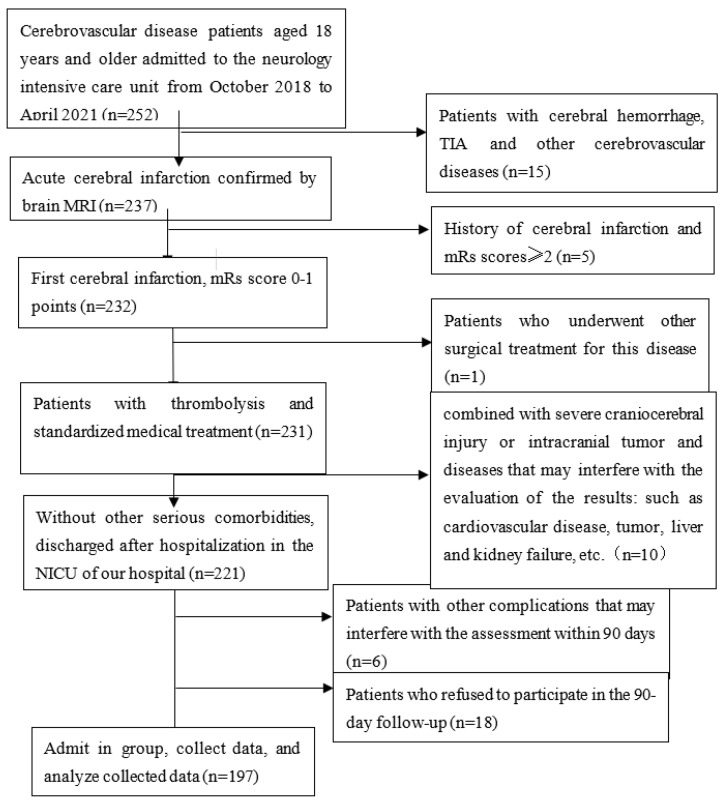
Details on study recruitment. mRS: modified Rankin scale; TIA: transient ischemic attack; MRI: magnetic resonance imaging; NICU: neurological intensive care unit.

**Figure 2 brainsci-13-00297-f002:**
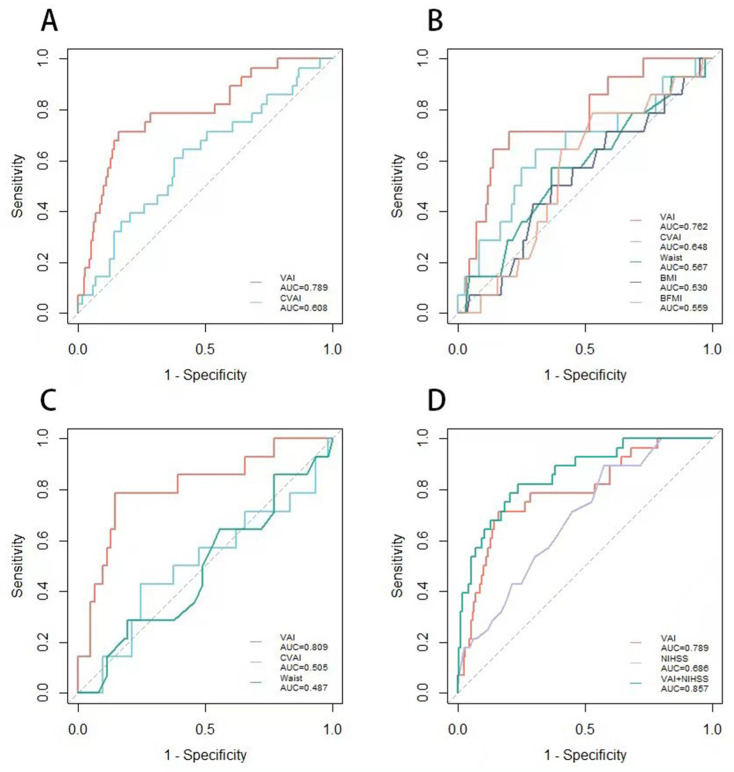
Area under the curves of VAI and other visceral adiposity markers for death after 90-day follow-up. (**A**) ROC curves of VAI, CVAI, and 90-day death among all patients. The cutoff value of VAI for 90-day death was 2.355 (sensitivity, 71%; specificity, 84%; area under the curve, 0.789; 95% confidence interval, 0.695–0.883). (**B**) ROC curves of VAI, CVAI, WC, BMI, BFMI, and 90-day death among male patients. The cutoff value of VAI for 90-day death was 2.202 (sensitivity, 71%; specificity, 80%; area under the curve, 0.762; 95% confidence interval, 0.631–0.893). (**C**) ROC curves of VAI, CVAI, WC, and 90-day death among female patients. The cutoff value of VAI for 90-day death was 2.765 (sensitivity, 79%; specificity, 85%; area under the curve, 0.809; 95% confidence interval, 0.675–0.943). (**D**) ROC curves of VAI, NIHSS, VAI + NIHSS, and 90-day death among all patients.

**Figure 3 brainsci-13-00297-f003:**
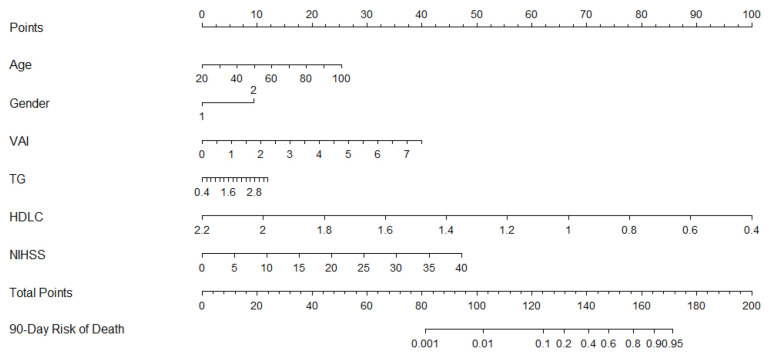
Nomogam for predicting death after 90-day follow-up.

**Table 1 brainsci-13-00297-t001:** Clinical baseline characteristic of VAI tertiles according to sex. Measurement data were expressed as mean ± standard deviation (x ± s), using the *t* test. Categorical data were expressed by assignment (%), using the Chi-square test or Fisher exact test. *: *p* < 0.005, **: *p* < 1.00 × 10^−6^.

Characteristics	Men				Women			
	VAI Tertiles1 (<1.12)	2 (1.12–1.90)	3 (>1.90)	*p*-Value	VAI Tertiles1 (<1.55)	2 (1.55–2.40)	3 (>2.40)	*p*-Value
**Number**	41	41	40		25	25	25	
**Age (years)**	70.73 ± 9.86	70.98 ± 12.71	72.15 ± 16.31	0.875	74.12 ± 10.50	65.88 ± 13.87	71.64 ± 11.25	0.05
**WC (cm)**	75.85 ± 7.61	78.37 ± 8.01	77.86 ± 9.16	0.351	71.20 ± 9.82	78.04 ± 11.37	80.80 ± 10.41	0.006 *
**BMI (m/kg^2^)**	22.25 ± 3.21	23.25 ± 3.23	23.43 ± 3.08	0.201	23.85 ± 4.50	23.34 ± 4.23	21.71 ± 3.78	0.174
**BFMI**	6.76 ± 2.15	7.49 ± 2.56	7.62 ± 1.86	0.171	10.22 ± 2.74	9.78 ± 2.48	8.06 ± 2.14	0.007 *
**Hypertension, *n* (%)**	31 (75.6%)	31 (75.6%)	30 (75.0%)	0.997	20 (80.0%)	20 (80.0%)	14 (56.0%)	0.092
**DM, *n* (%)**	10 (24.4%)	14 (34.1%)	15 (37.5%)	0.420	5 (20.0%)	5 (20.0%)	12 (48.0%)	0.043 *
**AF, *n* (%)**	12 (29.3%)	6 (14.6%)	5 (12.5%)	0.129	6 (24.0%)	9 (36.0%)	4 (16.0%)	0.262
**CHD, *n* (%)**	6 (14.6%)	5 (12.2%)	3 (7.5%)	0.540	2 (8.0%)	2 (8.0%)	2 (8.0%)	0.571
**Smoker, *n* (%)**	6 (14.6%)	5 (12.2%)	7 (17.5%)	0.569	0 (0%)	0 (0%)	0 (0%)	-
**NIH Stroke Scale**	13.66 ± 6.98	13.90 ± 9.05	10.33 ± 7.71	0.083	13.80 ± 7.12	15.32 ± 9.91	12.96 ± 4.94	0.541
**Systolic Blood Pressure (mmHg)**	154.41 ± 18.12	144.07 ± 20.53	143.45 ± 20.12	0.020 *	146.20 ± 22.04	146.92 ± 24.50	143.04 ± 24.56	0.827
**Diastolic Blood Pressure (mmHg)**	82.39 ± 15.00	81.83 ± 12.24	80.25 ± 15.31	0.783	79.84 ± 12.62	79.80 ± 9.38	76.88 ± 11.93	0.577
**HDL-C (mmol/L)**	1.23 ± 0.28	0.92 ± 0.21	0.76 ± 0.18	0.000 **	1.30 ± 0.28	1.00 ± 1.46	0.90 ± 0.18	0.000 **
**LDL-C (mmol/L)**	2.30 ± 0.82	3.00 ± 1.34	2.73 ± 1.14	0.021 *	2.34 ± 0.79	3.00 ± 1.03	2.73 ± 0.91	0.043 *
**Triglyceride (mmol/L)**	0.84 ± 0.20	1.13 ± 0.23	1.75 ± 0.51	0.000 **	0.88 ± 0.20	1.14 ± 0.25	1.64 ± 0.48	0.000 **
**Total Cholesterol (mmol/L)**	3.93 ± 1.00	4.42 ± 1.44	4.21 ± 1.35	0.232	4.17 ± 0.95	4.49 ± 1.02	4.36 ± 1.22	0.576
**Creatinine (μmol/L)**	104.66 ± 55.26	72.94 ± 29.32	73.07 ± 29.01	0.212	62.40 ± 24.68	69.62 ± 37.32	60.58 ± 20.90	0.499
**Uric Acid (μmol/L)**	312.02 ± 162.91	297.73 ± 130.29	291.58 ± 101.16	0.780	256.72 ± 110.76	268.80 ± 127.89	235.80 ± 103.57	0.590
**Fasting Blood Glucose (μmol/L)**	8.72 ± 4.18	7.30 ± 3.00	6.71 ± 2.43	0.554	6.69 ± 2.73	7.39 ± 2.75	8.34 ± 3.34	0.148
**Homocysteine (μmol/L)**	15.95 ± 7.33	17.38 ± 14.32	13.56 ± 5.83	0.292	21.31 ± 3.88	12.52 ± 3.79	13.54 ± 7.54	0.380
**Fibrinogen (g/L)**	4.31 ± 1.64	3.91 ± 1.99	4.51 ± 3.09	0.510	3.38 ± 1.45	3.90 ± 1.17	3.59 ± 1.25	0.387
**Hypersensitive C-reactive Protein (mg/L)**	13.22 ± 4.29	12.99 ± 4.45	11.89 ± 5.23	0.397	10.68 ± 5.07	12.52 ± 3.79	13.54 ± 7.54	0.459
**ALT (U/L)**	23.37 ± 19.74	26.81 ± 50.99	41.30 ± 87.89	0.359	20.32 ± 15.92	16.51 ± 9.16	23.49 ± 22.72	0.347
**AST (U/L)**	30.32 ± 14.64	30.02 ± 28.50	47.69 ± 126.80	0.481	25.52 ± 11.63	24.62 ± 14.24	27.00 ± 13.17	0.810
**Total Bilirubin (μmol/L)**	21.35 ± 12.97	19.26 ± 8.96	16.66 ± 9.05	0.136	18.21 ± 10.59	15.70 ± 9.83	15.09 ± 8.11	0.477
**Prealbumin (g/L)**	178.77 ± 55.66	179.06 ± 58.82	205.34 ± 71.76	0.094	169.81 ± 58.16	172.08 ± 51.97	196.39 ± 56.35	0.179
**Albumin (g/L)**	37.97 ± 5.01	36.95 ± 6.16	36.40 ± 5.58	0.444	38.13 ± 5.38	36.63 ± 6.71	35.90 ± 4.01	0.345
**WBC (10^9^/L)**	11.25 ± 4.38	9.72 ± 2.89	10.04 ± 4.31	0.185	8.30 ± 3.57	9.60 ± 3.62	9.01 ± 2.85	0.395
**Hemoglobin A1c (%)**	6.62 ± 1.60	6.98 ± 2.00	6.88 ± 1.64	0.669	6.50 ± 1.37	6.83 ± 2.33	15.04 ± 2.78	0.175
**TOAST Classification**	21/11/8/1	27/4/7/2	26/3/10/1	0.226	11/7/6/0	16/6/3/0	15/4/5/1	0.570
**Thrombolytic, *n* (%)**	5 (12.2%)	7 (17.1%)	8 (20.0%)	0.631	9 (36.0%)	3 (12.0%)	5 (20.0%)	0.119
**Embolectomy, *n* (%)**	4 (9.8%)	7 (17.1%)	2 (5.0%)	0.207	2 (8.0%)	1 (4.0%)	4 (16.0%)	0.332
**Intracerebral Hemorrhage, *n* (%)**	9 (21.95%)	5 (12.20%)	4 (10.0%)	0.285	6 (24.00%)	3 (12.00%)	2 (8.0%)	0.206

VAI: visceral adiposity index, WC: waist circumference, BMI: body mass index, BFMI: body fat mass index, DM: diabetes mellitus, AF: atrial fibrillation, CHD: coronary heart disease, NIH Stroke Scale: National Institutes of Health Stroke Scale, HDL-C: high-density lipoprotein cholesterol, LDL-C: low-density lipoprotein cholesterol, ALT: alanine aminotransferase, AST: aspartate aminotransferase, WBC: white blood cell, TOAST Classification refers to five classifications: (1) large-artery atherosclerosis, (2) cardioembolism, (3) small-vessel occlusion, (4) stroke of another determined etiology, and (5) stroke of an undetermined etiology.

**Table 2 brainsci-13-00297-t002:** Clinical baseline characteristics of patients with and without death after 90-day follow-up. Continuous data are shown as mean ± SD, values for dead and nondead patients were statistically significant according to a two-sample *t* test. Categorical data differences in patients between the two groups are represented with statistical significance that is based on the Chi-square test (χ^2^ and *p*) or Fisher exact test (Z and *p*). *: *p* < 0.005, **: *p* < 1.00 × 10^−6^.

Characteristics	Death after 90-Day Follow-Up (*n =* 28)	Without Death After 90-Day Follow-Up (*n =* 169)	*t*/χ^2^/Z	*p*-Value
**Gender male, *n* (%)**	14 (50%)	108 (63.9%)	*t =* 1.824	0.076
**Age (years)**	74.79 ± 11.68	70.37 ± 12.87	χ^2^ *=* 1.970	0.160
**WC (cm)**	77.41 ± 10.32	77.05 ± 9.35	*t =* 0.188	0.851
**BMI (m/kg^2^)**	22.36 ± 3.22	23.07 ± 3.67	*t =* −0.962	0.337
**BFMI**	7.88 ± 1.77	8.10 ± 2.69	*t =* −0.574	0.569
**Hypertension, *n* (%)**	18 (64.3%)	128 (75.7%)	χ^2^ *=* 1.642	0.200
**DM, *n* (%)**	7 (25.0%)	54 (32.0%)	χ^2^ *=* 0.543	0.461
**AF, *n* (%)**	8 (28.6%)	34 (20.1%)	χ^2^ *=* 1.328	0.515
**CHD, *n* (%)**	1 (3.6%)	21 (12.4%)	z *=* −1.377	0.168
**Smoker, *n* (%)**	3 (10.7%)	15 (8.9%)	χ^2^ *=* 0.098	0.754
**NIH Stroke Scale**	18.18 ± 9.21	12.34 ± 7.35	*t =* 3.746	0.000 **
**Systolic Blood Pressure (mmHg)**	141.07 ± 21.04	147.51 ± 21.34	*t =* −1.483	0.140
**Diastolic Blood Pressure (mmHg)**	77.36 ± 12.85	81.01 ± 13.20	t *=* −1.360	0.176
**HDL-C (mmol/L)**	0.78 ± 0.18	1.05 ± 0.29	*t =* −6.590	0.000 **
**LDL-C (mmol/L)**	2.70 ± 0.96	2.70 ± 1.09	*t =* 0.111	0.912
**Triglyceride (mmol/L)**	1.43 ± 0.50	1.20 ± 0.48	*t =* 2.348	0.020 *
**Total Cholesterol (mmol/L)**	4.04 ± 0.98	4.28 ± 1.24	*t =* −0.949	0.344
**Creatinine (μmol/L)**	59.80 ± 19.35	78.96 ± 81.54	*t =* −1.235	0.218
**Uric Acid (μmol/L)**	244.63 ± 143.11	289.03 ± 124.60	*t =* −1.709	0.089
**Fasting Glucose (μmol/L)**	7.51 ± 3.18	7.55 ± 7.36	*t =* −0.025	0.980
**Homocysteine (μmol/L)**	14.52 ± 7.17	16.04 ± 17.45	*t =* −0.410	0.683
**Fibrinogen (g/L)**	4.44 ± 3.30	3.94 ± 1.71	*t =* 1.156	0.249
**Hypersensitive C-reactive Protein (mg/L)**	10.65 ± 5.35	12.07 ± 4.91	*t =* −1.401	0.163
**ALT (U/L)**	38.90 ± 62.56	31.27 ± 62.64	*t =* 1.175	0.249
**AST (U/L)**	36.61 ± 35.27	31.27 ± 62.63	*t =* 0.439	0.661
**Total Bilirubin (μmol/L)**	15.08 ± 6.19	18.55 ± 10.73	*t =* −1660	0.098
**Prealbumin (g/L)**	190.25 ± 63.21	183.52 ± 60.21	*t =* 0.544	0.587
**Albumin (g/L)**	35.47 ± 5.34	37.28 ± 5.55	*t =* −1.608	0.109
**WBC (10^9^/L)**	8.93 ± 3.26	9.96 ± 3.86	*t =* 0.443	0.185
**Hemoglobin A1c (%)**	11.40 ± 24.78	7.35 ± 5.83	*t =* 0.781	0.443
**TOAST Classification †**	19/6/3/0	97/29/36/5	χ^2^ *=* 2.835	0.415
**Thrombolytic, *n* (%)**	5 (17.9%)	32 (18.9%)	χ^2^ *=* 0.018	0.892
**Embolectomy, *n* (%)**	2 (7.1%)	18 (10.7%)	χ^2^ *=* 0.324	0.569
**Intracerebral Hemorrhage, *n* (%)**	1 (3.6%)	28 (16.6%)	z *=* −1.793	0.073
**VAI**	2.93 ± 1.35	1.73 ± 0.92	*t =* 5.945	0.000 **
**CVAI**	99.10 ± 35.58	85.67 ± 34.88	*t =* 1.882	0.061

WC: waist circumference, BMI: body mass index, BFMI: body fat mass index, DM: diabetes mellitus, AF: atrial fibrillation, CHD: coronary heart disease, NIH Stroke Scale: National Institutes of Health Stroke Scale, HDL-C: high-density lipoprotein cholesterol, LDL-C: low-density lipoprotein cholesterol, ALT: alanine aminotransferase, AST: aspartate aminotransferase, WBC: white blood cell, † TOAST Classification refers to five classifications: (1) large-artery atherosclerosis, (2) cardioembolism, (3) small-vessel occlusion, (4) stroke of another determined etiology, and (5) stroke of an undetermined etiology. VAI: visceral adiposity index, CVAI: Chinese visceral adiposity index.

**Table 3 brainsci-13-00297-t003:** Area under the curves of VAI and other visceral adiposity markers for death after 90-day follow-up. WC: waist circumference, BMI: body mass index, BFMI: body fat mass index, VAI: visceral adiposity index, CVAI: Chinese visceral adiposity index, AUC: area under the curve.

Markers	All (*n =* 197)		Men (*n =* 122)		Women (m *=* 75)	
	AUC	95% CI	AUC	95% CI	AUC	95% CI
**WC (cm)**	0.520	0.401–0.640	0.567	0.405–0.729	0.487	0.318–0.655
**BMI (m/kg^2^)**	0.445	0.334–0.556	0.530	0.374–0.687	0.484	0.230–0.538
**BFMI**	0.487	0.389–0.585	0.559	0.416–0.702	0.328	0.187–0.470
**CVAI**	0.608	0.494–0.723	0.648	0.483–0.813	0.505	0.325–0.684
**VAI**	0.789	0.695–0.883	0.762	0.631–0.893	0.809	0.675–0.943

**Table 4 brainsci-13-00297-t004:** Multivariable logistic regression model for predicting death after 90-day follow-up. TG: Triglyceride, NIHSS: National Institutes of Health Stroke Scale, VAI: visceral adiposity index.

Variables	Odds Ratio	95% CI	*t*	*p*-Value
**Age**	1.034	0.992–1.077	0.033	0.111
**Gender**	1.868	0.625–5.587	0.625	0.263
**TG**	0.236	0.047–1.182	−1.444	0.079
**NIHSS**	1.136	1.068–1.207	0.127	0.000
**VAI**	5.944	2.752–12.837	1.782	0.000

## Data Availability

The datasets generated during the current study are available from the corresponding author on reasonable request.

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
