# Peer review of "Derivation and Validation of a New Visceral Adiposity Index for Predicting Short-Term Mortality of Patients with Acute Ischemic Stroke in a Chinese Population"

_brainsci, 2023, doi:10.3390/brainsci13020297_

Round 1

Reviewer 1 Report

The authors present the results of a single-center retrospective cohort study of 197 patients with acute ischemic stroke to analyze the association of obesity and metabolic related indicators, including visceral adiposity index (VAI), BMI, and body fat mass index with short-term mortality. In the study, the authors found that VAI correlated positively with 90-day death, and that this factor could be used as an independent risk factor to predict death in acute ischemic stroke. Its predictive value was significantly higher to that of traditional obesity indicators, such as BMI, WC and BFMI. The study is potentially interesting, but some aspects of the manuscript may be improved taking into account the following points:   

        1. In the Abstract, please add in the line 29 the traditional metabolic indicators. 
        2. It would be interesting to describe the causes of death (neurological and non-neurological) in the study sample. 
        3. It would be interesting to know the different ischemic stroke subtypes in the study population. 
        4. Please avoid “etc.” In Discussion (line 328) 
        5. Typographic errors in Discussion  (“morality”; -line 337-; “inducesbrain” –line 348 -) should be corrected. 
        6. The authors should mention that an indispensable line of research in the future would be precisely the assessment of the visceral adiposity index to predict mortality in acute lacunar versus non-lacunar ischemic stroke, since the pathophysiology, prognosis and clinical features of acute small-vessel ischemic strokes are different from other types of cerebral infarcts; and lacunar infarcts are the stroke subtype with the best functional prognosis (see and add this recent reference: Int J Mol Sci 2022; 23, 1497).    

Author Response

Dear reviewer:

Thank you for your comments and suggestion concerning our manuscript. The
comments and suggestions are all valuable and very helpful for revising and improving our paper, as well as the important guiding significance to our researches. We have studied comments carefully and have made correction which we hope meet with approval.

The response to your comments has been upload as Word file below.

Yours sincerely

Xiang Tang

Reviewer 2 Report

In the Abstract, lines 28-29

"Patients with VAI 2.355 had a higher risk of short-term mortality VAI has a predictive value higher than that of traditional metabolic indicators such as." 

the sentence does not continue, please modify it.

Please write the keywords in lowercase.

Add the reference to the sentences: "At present, there are few studies on the correlation between VAI and the  prognosis of patients with cerebral infarction." and "The relationship between overweight, 67 obesity, and metabolic disorders and the short-term prognosis of patients with acute cerebral infarction also remains controversial."

Overall, the introduction section could be divided into more sections, with additional information about the subject because you have chosen a very wide and interesting theme to study.

In the discussion session, more information could be added, as a comparison between your study and other studies on the same subject. 

Also, please add more valuable conclusions. 

Author Response

(The authors gave the same response as above.)

Round 2

Reviewer 1 Report

I appreciate the authors efforts to include my comments. I do not have further comments.

Reviewer 2 Report

Congrats!